# Comparison to Micro Wear Mechanism of PVD Chromium Coatings and Electroplated Hard Chromium

**DOI:** 10.3390/ma16072695

**Published:** 2023-03-28

**Authors:** Zhongyi Yang, Ning Zhang, Hongtao Li, Bo Chen, Bo Yang

**Affiliations:** 1College of Materials Science and Engineering, Nanjing Tech University, Nanjing 211800, China; 202061103116@njtech.edu.cn (Z.Y.); 202061203134@njtech.edu.cn (N.Z.); 202161203196@njtech.edu.cn (B.C.); 2Jiangyin Innovation Institute of Metal Materials Co., Ltd., Wuxi 214433, China; 13537758248@163.com

**Keywords:** electroplated hard chromium, physical vapor deposition, tribological behavior, chromium coatings

## Abstract

Electroplated hard chromium (EPHC) has been widely used in industry due to its excellent mechanical properties, but the development of this technology is limited by environmental risks. The physical vapor deposition (PVD) process has shown promise as an alternative to EPHC for producing chromium-based coatings. In this research, we investigate the microstructure and wear resistance of pure chromium coatings using two PVD techniques, namely, magnetron sputtering ion plating (MSIP) and micro-arc ion plating (MAIP), which are compared to EPHC. To assess wear resistance, we evaluated factors such as hardness, coating base bonding force, wear rate and friction coefficient via friction and wear experiments. The results show that, in terms of microstructure, while the EPHC coating does not exhibit a strong preferred growth orientation, the PVD coatings exhibit an obvious preferred growth orientation along the (110) direction. The average grain size of the EPHC coating is the smallest, and the PVD chromium coatings show a higher hardness than the EPHC coating. The results of pin-on-disk tests show that there is little difference in friction coefficients between EPHC and MAIP chromium plating; however, the MAIP chromium coating showed an excellent specific wear rate (as low as 1.477 × 10^−13^ m^3^/Nm). The wear condition of the MAIP chromium coating is more stable than that of the EPHC coating, indicating its potential as a replacement for EPHC.

## 1. Introduction

The properties and characteristics of parts surfaces are very important for a wide variety of applications. The surface properties, combined with the bulk material properties, give parts unique behaviors [1]. Chromium plating is a widely used technique across various industries and can be classified into two types: decorative chromium and hard chromium electroplating (EPHC). Decorative chromium typically has a thickness of approximately 0.25 μm, while EPHC is a functional coating with a maximum thickness of 500 μm [2]. It can generally reach tens of microns and is widely used in the field of wear and corrosion resistance to protect parts such as spinning cups, steel collars and other high-speed sliding parts [3,4,5]. However, EPHC’s production process poses significant environmental risks, specifically, the generation of hazardous Cr (VI) compounds. Due to the high toxicity of Cr (VI) compounds, the European Regulation (EC) No. 1907/2006 (“reach”) has restricted or prohibited the use of chromium trioxide baths for functional and decorative chromium plating since 1 September 2017. There is an urgent need to develop alternative processes to replace hexavalent chromium plating [6,7,8].

Nowadays, various research studies are underway to explore alternative processes to hexavalent hard chromium plating. These include trivalent chromium, thermal spraying, chemical vapor deposition (CVD), and physical vapor deposition (PVD). Because trivalent chromium has low toxicity and low pollution, it is regarded as one of the alternatives to hexavalent hard chromium plating with the most potential [9]. However, it is difficult to thicken trivalent chromium in actual production, as the plating solution can be unstable. As a result, there exist difficulties in the cost of research and development and in application products, and trivalent chromium is subject to many restrictions in practical application [10,11,12,13]. CVD generally needs to be between 900 °C and 1100 °C, and the high-temperature tempering of general steel occurs between 500 °C and 600 °C. There are defects in the preparation and production of steel-based coating, and the actual material selection range is small [13]. Among the alternative processes explored, Cr-based coatings produced using physical vapor deposition (PVD) have shown promising mechanical properties and wear resistance [14,15,16]. Magnetron sputtering ion plating (MSIP) uses gas ionization to spit out particles and then deposit them on the substrate to obtain a compact and uniform coating. It has been used to produce chromium coatings with microhardness comparable to EPHC [17]. However, due to the stress and low deposition rate of the coating, it is difficult to thicken and improve its wear resistance; thus, it has not reached complete commercial maturity [18,19]. Recently, new techniques in the field of magnetron sputtering have emerged, and ongoing research aims to address the industrialization challenges of this technology [20]. One such technique is micro-arc ion plating (MAIP). By rearranging the internal magnetic field and changing the discharge area of the target to improve its current density, the discharge enters the glow-arc discharge transition zone. Compared with MSIP, MAIP has higher current density bombarding the target, which makes the plating ions have higher kinetic energy and a higher ionization rate, and improves the density and deposition efficiency of the coating [21,22].

In various industries, wear is the most common cause of the failure of substrates and coatings, including EPHC. Wear leads to material transfer and local warping of the material from the surface, which prevents sliding to other surfaces [23]. The microhardness, surface friction coefficient and average grain size of the material all have a certain effect on the wear resistance, and also affect the wear failure mode of the coating [24,25,26]. Coatings prepared by EPHC and PVD techniques have different microstructures and mechanical properties that affect their wear resistance [27,28]. So far, the current research on hard chromium replacement electroplating mainly focuses on replacement coatings other than pure chromium, and there is limited research on the tribological properties of pure chromium coatings deposited by PVD. In this study, based on the characteristics of the two kinds of ion plating mentioned above, we adopted the pure chromium coatings deposited by the two types of ion plating and the hard chromium coating and analyzed the difference in the wear resistance of chromium coatings caused by the difference in the microstructure and caused by the difference in the off-target mechanism of the two kinds of ion plating particles. In addition, the difference in the mechanical properties and wear resistance mechanisms of the three kinds of chromium coatings were investigated.

## 2. Materials and Methods

### 2.1. Coating Deposition

EPHC was obtained from the supplier, PVD chromium coatings were prepared by MSIP019 closed field unbalanced magnetron sputter ion plating and the target material distribution diagram is shown in Figure 1. The MSIP chromium coating and the MAIP chromium coating are made of high-purity chromium target materials and are 300 mm × 100 mm and φ100 mm (purity is 99.99%), respectively. These coatings were deposited on GCr15 bearing steel, a commonly used base material for steel collars. The sample size used in the experiment was φ40 mm × 4 mm, and its surface was ground, polished and washed with acetone and alcohol using an ultrasound for 15 min and then dried with nitrogen before deposition. The gas used in the experiment was 99.99% high-purity Ar, and the background vacuum of sputtering was lower than 5.0 × 10^−5^ Torr. In the experiment, when the MSIP chromium coating and the MAIP chromium coating were used, the apparent mass flow rate of argon during sputtering was controlled by mass flowmeter to 18 sccm and 10 sccm, respectively. The deposition times were 120 min and 60 min for MSIP and MAIP, respectively, with a distance of about 120 mm between the magnetron target and the sample holder. The specific parameters have been presented in Table 1.

### 2.2. Microscopic Characterization

X-ray diffraction (XRD) can conduct qualitative and quantitative analyses of the crystal structure, phase composition and stress of the sample materials. In this experiment, D/Max-2400 type X-ray diffraction (XRD, Cu Kα radiation) was used to analyze the phase composition, average grain size and crystal orientation of the pure chromium coatings prepared by the test. Because of the thin coatings, the test was conducted with a small Angle grazing, the scanning range was 30~90°, the step length was 0.02°, the scanning speed was 4 °/min and the grazing Angle was 1°. Grain size was estimated using the Scherrer Equation (1):(1)$$\tau =\frac{0.94\lambda }{\beta cos\theta },$$
where τ is the average grain size, β is the half-maximum peak width after instrument broadening correction, θ is the Bragg Angle, and λ is the X-ray wavelength [29].

The pure chromium coatings on the steel plate samples were cut, cold-set, pre-ground and polished. The microscopic surface and cross-section morphology of the pure chromium coatings were observed and analyzed by a field emission scanning electron microscope (SEM, JSM-IT500, Tokyo, Japan). The thickness of the coating was measured while the cross-section morphology of the coating was observed, and then the deposition rate of the pure chromium coating was calculated to analyze the influence of the deposition mode on the coating thickness. A metallographic microscope (DMM-400C, Beijing, China) was used to observe and analyze the morphology of indentation.

### 2.3. Mechanical and Tribological Properties

The microhardness of the coating was measured directly using an HVS-1000 Vickers microhardness tester (CSM instrument microcombination tester, China). The test involved a 100 g load and a 10 s pressure holding time. The microhardness of the steel sheet coating sample was measured at 5 different points, and the average value was used. The toughness of the coating was qualitatively compared using the indentation method. The test load was 100 g and the pressure holding time was 10 s. The coating base adhesion of the chromium coating was tested with a WS-2005 coating adhesion automatic scratching instrument. The maximum load was 100 N, with a loading speed of 100 N/min. When the coating was peeled off for the first time in the scratch, it indicated that the membrane base interface began to fail. When a large amount of the coating was peeled off from the substrate in the scratch, the load at this time was the maximum critical load Lc. In this paper, the critical load Lc, when the coating completely fails, is obtained by combining the acoustic emission signal of the scratch meter and the measurement and calculation in the scratch topography. The bonding strength of the coating is then evaluated.

A laboratory homemade ball-disc friction and wear tester (XLGT200, Xi’an, China) was used to measure the tribological properties of the coating. The loading weight was applied to a small ball fixed at the front end of the loading rod, which served as the friction pair. GCr15 balls with a diameter of 5 mm were used as the friction pair material for this experiment, and were slid on the sample for 30 min with a load of 4 N and a speed of 400 r/min. The friction ring had a radius of 24 mm, and a computer recorded the friction coefficient. At the same time, the weight of samples before and after the friction and wear test was measured to calculate the weight loss and volume wear rate.

## 3. Results

### 3.1. Characteristics of Microstructure

#### 3.1.1. X-ray Diffraction

The body-centered cubic structure (BCC) is exhibited by all three chromium coatings. According to the calculation of the texture coefficient (TC) of different crystal orientations, it is possible to derive the diverse favored growth orientations of the chromium coatings. The XRD of the three chromium coatings is shown in Figure 2. In the XRD analysis of the three chromium coatings, the ICDD is 85-1336-64712. The EPHC coating’s crystal peak is wider and its relative strength is lower than that of the MSIP chromium coating, indicating the grain size and/or micro-strain within the grain. According to the Scherer formula, the average grain sizes of the MAIP, MSIP and EPHC coatings were found to be 12.1 nm, 20.2 nm and 13.3 nm, respectively. The average grain of the MAIP chromium coating is small. This is because the MAIP combines the “cascading elastic collision” miss mechanism of the magnetron sputtering technology and the “thermal emission miss” of the multi-arc ion plating to form a mixed miss mechanism of “collision miss + emission miss” [20]. The power density of the target surface was improved by increasing the target voltage, resulting in high-density sputtering deposition, which made the surface of the MAIP chromium coating denser and finer than the surface of the MSIP chromium coating, and consequently, the grain structure of the MAIP chromium coating was found to be finer.

#### 3.1.2. SEM

The deposition thicknesses and deposition rates of the three chromium coatings in Figure 3 are calculated by measuring the thickness of the coatings at 10 positions to obtain their average values. Figure 4 shows the surface and cross-section morphology of the chromium coatings produced by the three different techniques.The growth orientation of the EPHC crystals is inconsistent, but the deposition rate is high. In addition, the surfaces of the two PVD chromium coatings appear to be smooth and without noticeable surface imperfections, and no voids are observed between the crystal structures of the cross-sectional coatings.

Figure 4a,b show the surface and the cross-section of the EPHC coating with a thickness of 10.20 μm, which exhibit a honeycomb structure with the irregular stacking of grains on the microsurface. This is caused by stress accumulation, which ultimately leads to the formation of a crisscross mesh microcrack structure on the surface. The high deposition rate results in micron-sized thick plating, which combines with a microcrack structure to form an abrasion-resistant coating. Figure 4c,d show the surface and the section morphology of the MSIP Cr coating. Under the surface scanning electron microscope, it shows a clear arched granular structure and a dense columnar crystal structure. This is because chromium particles on the surface of the target are sputtered out and deposited on the substrate under high-energy bombardment. During the growth process, internal defects are continuously produced and the surface diffusion ability is constantly improved. Internal defects can provide periodic nucleation sites for Cr atoms deposited later and greater surface diffusivity may accelerate the growth of Cr nuclei along the Cr (110) crystal plane. Thus, a distinct, dense columnar structure can be formed without any voids [11,12]. Compared with the MSIP Cr coating, MAIP technology focuses higher sputtering power on the target and improves the sputtering efficiency of the coating, while also providing higher energy for such atomic review and diffusion. Therefore, the MAIP Cr coating can obtain the columnar crystal structure with a more compact surface particle arrangement and cross-section. Figure 4e,f show the surface and the section morphology of the MAIP Cr coating. In this case, the grain size of the MAIP chromium coating is smaller than that of the MSIP chromium coating, which is consistent with the XRD analysis results.

### 3.2. Tribological and Mechanical Properties Testing

#### 3.2.1. Mechanical Properties

After calculation, the intrinsic microhardness of chromium coatings is obtained. The surface microhardness of the three chromium coatings is shown in Figure 5. Compared with the EPHC with a microhardness of HV 734.74 ± 16.48, the surface microhardness of the two ionic coatings is slightly improved, the microhardness of the MSIP coating is HV 829.74 ± 14.80, and the microhardness of the MAIP coating is HV 994.66 ± 21.33.

Figure 6 exhibits the microhardness indentations of the three chromium coatings. As shown in Figure 6a, the EPHC coating produced large indentations with low microhardness due to the presence of more crack defects on its surface. The indentation on the upper left part of the EPHC coating appeared to have larger cracks after the application of the load, with some extending cracks. In Figure 6b, wavy cracks (typical ductile crack morphology) appeared on the four sides of the indentation, with non-significant outward spreading cracks appearing along the four corners of the indentation. The columnar crystals penetrate the matrix after being subjected to pressure, which may be the main reason for the cracking of the chromium coating. In Figure 6c, the indentation morphology of the MAIP chromium coating is small, and there are no obvious cracking and stripping phenomena around it. The study demonstrates that the plastic-modification resistance of the coating increases with the decrease in the grain size. A smaller grain size leads to more grain boundaries, and with the stronger ability to hinder dislocations, the microhardness increases [30].

Figure 7 exhibits the coating base bonding strength of the coating with the scratch morphology and acoustic emission signal. As shown in Figure 7a, in the whole scratch experiment, the EPHC coating has no obvious cracks and the coating base bonding strength is adequate with the combination of acoustic emission signals. The AE signal of the MSIP chromium coating appears for the first time at 16 N; the corresponding signal is L0. When the second time is called, and the large spalling is at 19 N, a strong signal is also generated, and the corresponding signal is Lc, as shown in Figure 7b. With the increase in signal strength, fragmentation is intensified. Furthermore, the maximum critical load of the MAIP chromium coating is 18 N.

#### 3.2.2. Galling Wear Testing

Figure 8 illustrates the friction and wear curves of the three chromium coatings, along with their wear rate. The friction and wear process of the EPHC coating is unstable. At the initial friction stage, the surface is easily polished into a slightly convex body without deformation, resulting in a low friction coefficient. However, the friction coefficient suddenly increases to 0.6 at 510 s. After surface wear, due to the higher thickness and microhardness of the coating, the friction wear remains in a rough and unstable stage. At this stage, a thickness of 2 μm has already worn out, and the wear distance per unit thickness is 452.16 m. The wear failure of the MSIP chromium coating is divided into three stages. The first stage (the initial 300 s) is the run-in wear stage, and the friction coefficient of the coating surface is about 0.35. This is because the surface is flat and the roughness is low. Then, with the increase in friction shear stress, the friction coefficient gradually increases to 0.75. The second stage is the stable wear stage (starting from 300 s), in which the wear is due to the high microhardness. However, the thickness is only 2 μm, resulting in the complete failure of the coating at 380 s, exposing the substrate. Thus, the limit of the service life is 380 s with a wear distance per unit thickness of 88.79 m. Differently from the MSIP chromium coating, which has an unstable friction and wear process with a high friction coefficient for a long time, the MAIP chromium coating in the process of friction and wear is more stable and the wear rate is lower. The friction and wear process consists of three processes, with the first stage being the run-in wear stage (at the beginning of 160 s). The second stage is the stable wear stage (starting after 160 s). The chromium coating has high hardness, but limited thickness, so the effective wear time (service life) of the coating is 560 s, with a wear distance per unit thickness of 179.20 m. During this time, the friction and wear curve does not change greatly. Compared with the MSIP chromium coating, the MAIP chromium coating enters the stable wear stage faster, and the friction coefficient is lower and the wear resistance time is longer.

It is well known that the wear mechanisms are commonly identified by the morphology of the wear marks, as well as the form and size of the wear fragments. Kovarikova et al. classified the typical wear phenomena caused by the major wear mechanisms as adhesion (scuffing or galling areas, holes, plastic shearing, material transfer), abrasion (scratches, grooves, ripples), fatigue of the surface (cracks, pitting) and reaction of the tribochemical (reaction products (layers, particles)) [6,31].

Figure 9 shows the surface wear morphologies of the three chromium coatings after the friction and wear tests. Figure 9a–c show the wear widths of the EPHC, MSIP and MAIP chromium coatings, respectively. In addition, the local wear morphology of the typical features of the three pure chromium coatings is analyzed microscopically, and the wear mechanism of different chromium coatings is studied, as shown in Figure 9d–i. Specifically, Figure 9d–i show local enlarged images of positions A and B in Figure 9a–c.

The total width of wear for the EPHC coating is 537 ± 11 μm. Figure 9d,g demonstrate its local wear morphology. It can be seen that chromium coating separation occurs during the process of friction and wear. The MSIP chromium coating exhibits an overall wear width of 650 ± 19 μm, and Figure 9e,h illustrate its local wear morphology, with the wear edge exhibiting serrate fracture. The coating is completely worn through and fails during the process of friction and wear, and the fracture position of the wear edge reveals the substrate and chromium coating to have an obvious metal grain shape. At the wear marks position, it is obvious that after the friction and wear test, the coating material is crushed to form part of a massive metal and granular structure adhering to the substrate surface, indicating a typical feature of abrasive wear [32]. The total width of wear for the MAIP chromium coating is 466 ± 12 μm. Figure 9f,i show the local wear morphology of the coating. The excessive area of the wear edge shows small saw tooth fluctuations. The local wear marks are rough, the coating materials are crushed in the adhesive part and the wear forms are typical abrasive and adhesive wear [26].

## 4. Discussion

In studying the wear behavior of coatings, it has been noted that the high hardness and wear resistance of EPHC have been the subject of considerable speculation and research over the years, having been attributed to various factors including the hydrogen and oxygen content of the film, small grain size and internal stresses [3,5]. The microstructure of the coating, the orientation and size of grain growth, the thickness of the coating, and the microhardness of the coating are proven to affect the wear resistance of the coating in this paper. In this study, we found that, compared with the MSIP coating, the surface structure of the MAIP coating is denser and smoother, and the difference in microstructure directly affects the microhardness of the coating. Compared with the MAIP coating, the surface roughness of the MSIP coating may reduce the wear resistance of the coating. In addition, it is widely accepted that coatings have maximum hardness at a critical grain size and that H significantly decreases with an increase in grain size (the Hall–Petch effect) [19]. Based on the analysis of the average grain size of the three chromium coatings, the microhardness of the MAIP chromium coating is higher than that of the MSIP chromium coating.

Although the microhardness of the PVD chromium coating is higher than that of the EPHC coating, the wear resistance of the PVD coating is worse than the EPHC’s. One of the reasons is that the difference in the microstructure of the chromium coatings leads to the difference in its surface microhardness and friction coefficient. In the friction and wear experiment, the abrasive particles of the coating caused by sliding contact and high contact stress are the main sources of wear. Another factor contributing to the difference in wear resistance is the thickness of the coating. However, we found that the specific wear rate of the MAIP chromium coating was as low as 1.477 × 10^−13^ m^3^/Nm, and the average wear performance per unit thickness of the coating (friction stability and wear distance) has the potential to replace EPHC in some tribological applications. Therefore, the relationship between the crystal microstructure of the MAIP chromium-based coatings and coating thickening can be further explored in subsequent studies.

## 5. Conclusions

In this study, the microstructure and tribology properties of the PVD chromium coatings were prepared and detected with respect to the EPHC as a the reference object. After comparing and analyzing the differences in the microstructure and tribology properties among the three chromium coatings, the following conclusions were drawn: It was found that the MAIP realized high-power sputtering deposition and refined grain organization through the “collision miss + emission miss” hybrid mechanism. Compared with the EPHC coating, the refinement of the average grain of the MAIP chromium coating improved the microhardness, and the wear rate of the coating was as low as 1.477 × 10^−13^ m^3^/Nm. After the calculations of the friction and wear experiments, it was found that the average wear resistance distances of the unit thickness of the EPHC, MSIP and MAIP chromium coatings are 88.79 m, 179.2 m, and 452.16 m, respectively. The wear resistance of the MAIP chromium coating is significantly higher than that of the MSIP coating, which has a wear resistance comparable with that of the EPHC coating. Furthermore, the wear process of PVD chromium coatings is mainly abrasive wear, which differs from the stripping wear and abrasive wear observed in EPHC.

## Figures and Tables

**Figure 1 materials-16-02695-f001:**
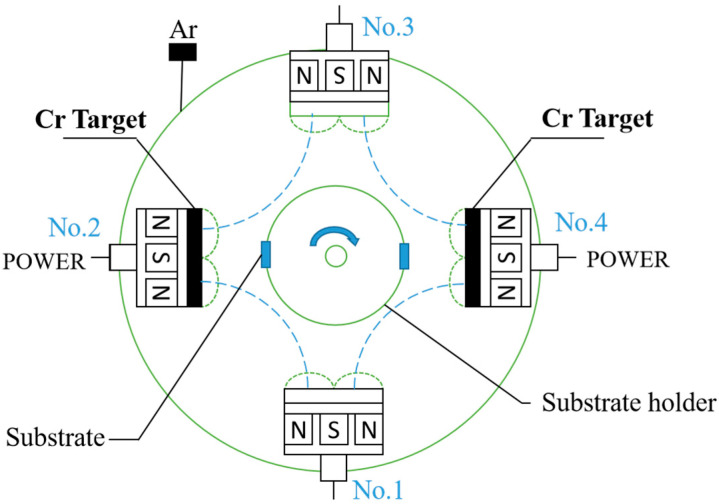
Deposition process of the chromium coatings and schematic diagram of vacuum chamber (top view).

**Figure 2 materials-16-02695-f002:**
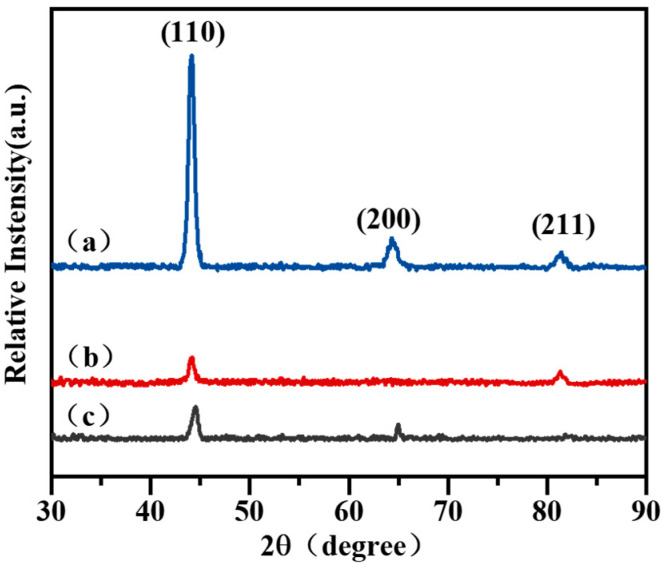
The XRD pattern of chromium coating surface is normalized with the peak of maximum strength, showing the difference in crystal orientation. (**a**) MAIP; (**b**) MSIP; (**c**) EPHC.

**Figure 3 materials-16-02695-f003:**
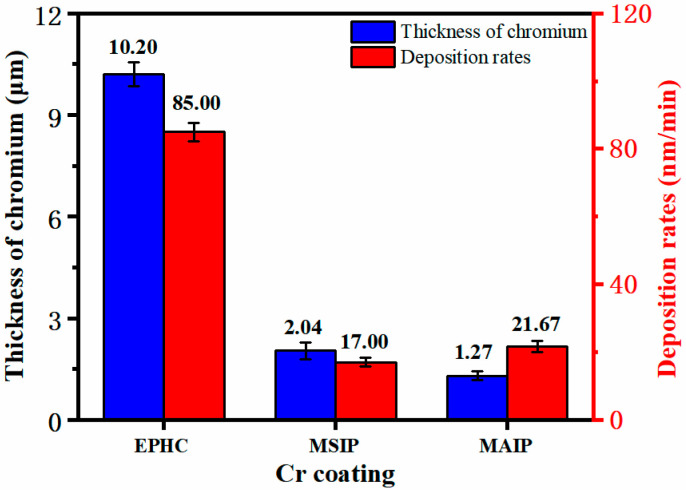
Thickness and deposition rates of chromium coatings deposited using EPHC, MSIP and MAIP.

**Figure 4 materials-16-02695-f004:**
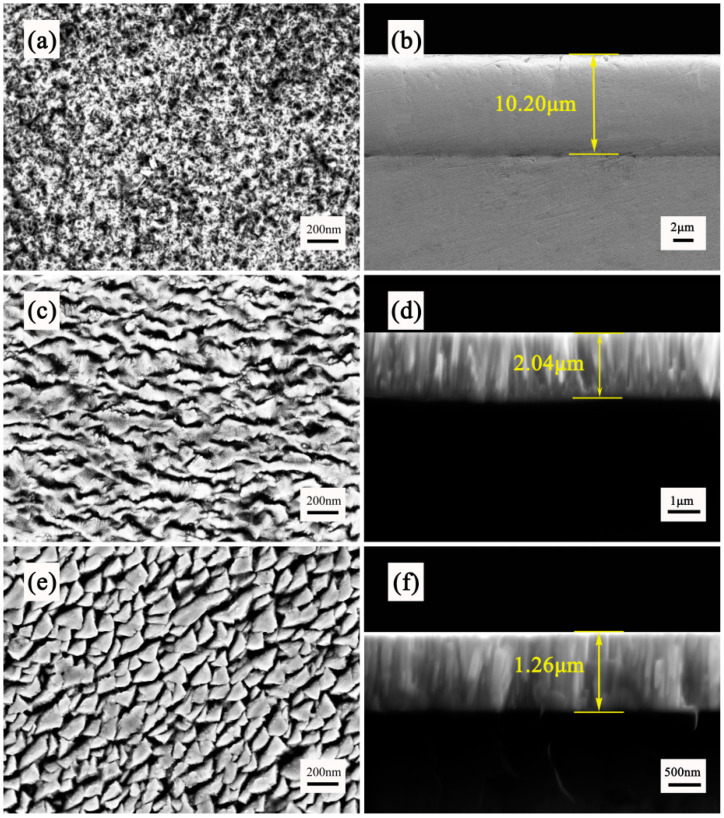
SEM micrographs of surface and cross-section of Cr coatings deposited under (**a**,**b**) EPHC; (**c**,**d**) MSIP; (**e**,**f**) MAIP.

**Figure 5 materials-16-02695-f005:**
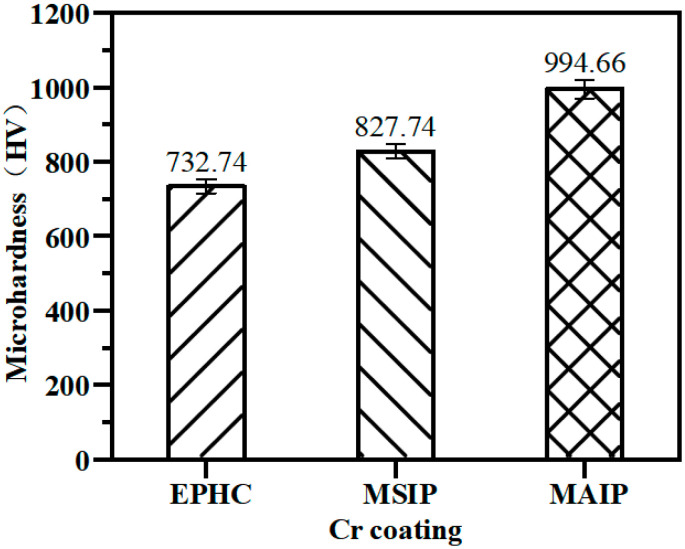
Mean mircohardness measured on the top surface of chromium coatings (working load 10 g, holding load 10 s).

**Figure 6 materials-16-02695-f006:**
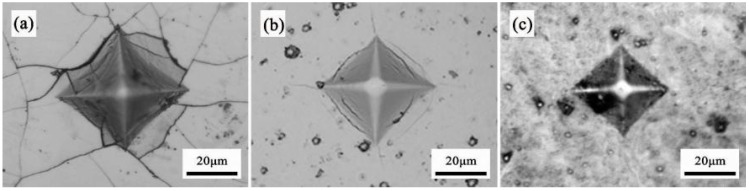
Microhardness morphology of chromium coatings (working load 100 g, holding load 10 s) (**a**) EPHC; (**b**) MSIP; (**c**) MAIP.

**Figure 7 materials-16-02695-f007:**
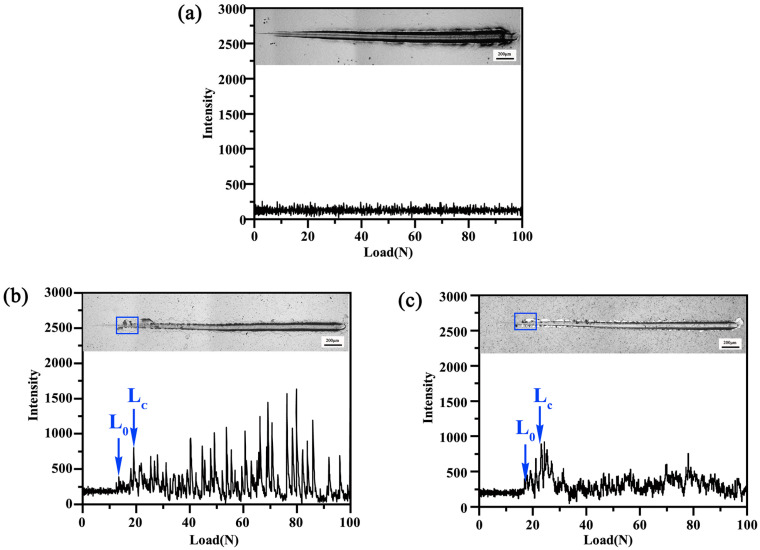
Surface scratch morphology and acoustic emission signals of three chromium coatings (load 100 N, loading time 60 s) (**a**) EPHC; (**b**) MSIP; (**c**) MAIP.

**Figure 8 materials-16-02695-f008:**
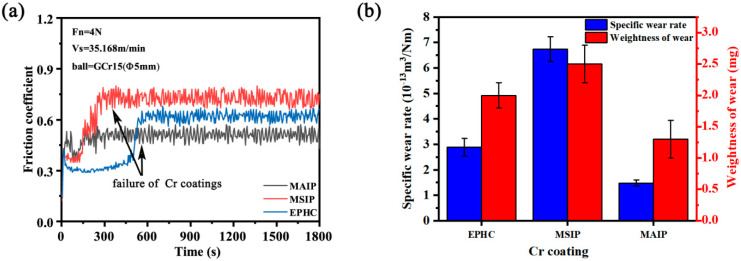
(**a**) The friction coefficient of the three chromium coatings; (**b**) specific wear rate and weight of wear of the three chromium coatings.

**Figure 9 materials-16-02695-f009:**
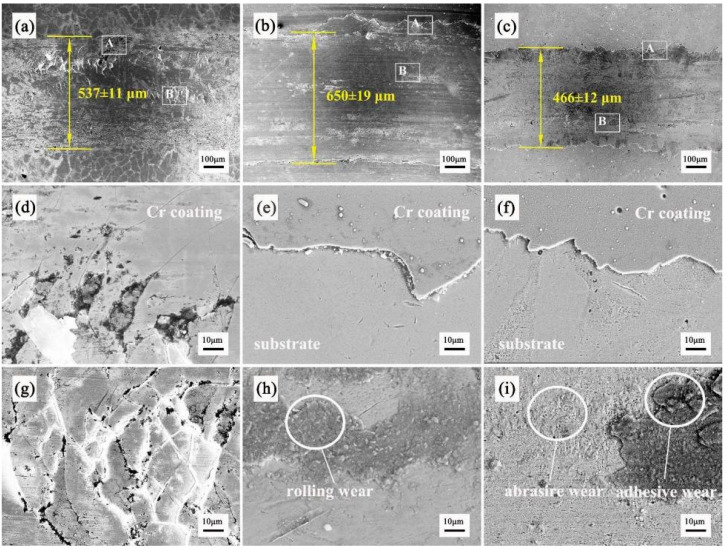
The amplification of surface wear morphology and the local wear morphology of the three pure Cr coatings (**a**,**d**,**g**), EPHC (**b**,**e**,**h**), MSIP (**c**,**f**,**g**,**i**) and MAIP (**d**,**g**,**e**,**h**), and (**f**,**i**) the local amplification of positions A and B (**a**–**c**).

**Table 1 materials-16-02695-t001:** Process parameters of the magnetron sputtering ion plating and the micro-arc ion chromium plating on GCr15.

Processing	MSIP	MAIP	Pulsed Bias		
	I_Cr_ (A)	I_Cr_ (A)	Voltage (−V)	Frequency (kHz)	Pulse Width (μs)
Ion cleaning	0.3	0.3	400	250	0.5
Cr layer	2.5	1.5	60	50	1.5

## Data Availability

Not applicable.

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
