# Peer review of "Comparison to Micro Wear Mechanism of PVD Chromium Coatings and Electroplated Hard Chromium"

_materials, 2023, doi:10.3390/ma16072695_

Round 1
Reviewer 1 Report
The article investigates the mechanical properties of Cr coatings produced by different methods, comparing their wear resistance and microstructure. The authors examined pure chromium coatings produced by magnetron sputtering ion plating (MSIP), micro-arc ion plating (MAIP), and electroplated hard chromium (EPHC). Through friction and wear experiments, the study found that the PVD chromium coatings had superior wear resistance and higher hardness than EPHC. The MAIP chromium coating exhibited the most stable wear condition and an excellent specific wear rate, suggesting that PVD techniques hold promise as an alternative to EPHC for producing chromium-based coatings.
The research outcomes are presented with scientific rigor, making it a valuable resource for a broad range of materials researchers and industry applications. The clarity and organization of the author’s presentation in this article make their insights easily accessible to researchers who are new to the field. However, there are a few minor recommendations that could help to further improve the manuscript.
· 1) Authors have compared Cr coating deposited using three different methods. However, to achieve the best outcomes of comparison, the thickness of each coating must be identical. Therefore, it is important to deposit all three coatings at the same rate of deposition and to the same thickness to make a meaningful comparison of their mechanical and wear resistance properties. Alternatively, the authors could justify the reason for using different thicknesses for the coatings in this study, to ensure that the comparison is still meaningful.
· 2) To improve the reliability of the indentation test results, the authors could consider calculating the critical (threshold) stress for each film and then applying an equal percentage of force to all films. This approach would account for the difference in thickness between the EPHC, MAIP, and MSIP films, and provide more accurate comparative data. Additionally, this approach could help identify the maximum stress that each film can withstand before cracking, which could be useful for future testing and development.
Reviewer 2 Report
The paper is verry interesting and in certain way may contribute to the journal. The paper is written in solid English and is easy to follow.
Author Response
point: The paper is verry interesting and in certain way may contribute to the journal. The paper is written in solid English and is easy to follow.
respond: Thank you very much for taking the time and effort to read my article, and thank you for your recognition of this article.Thank you very much.
Reviewer 3 Report
The paper "Comparison to micro wear mechanism of PVD chromium coatings and electroplated hard chromium" is suitable for publication in the Materials journal after some minor corrections:
1. The introduction is in general well written, but there are some aspects that need to be updated: the authors should add the influence of Cr in terms of mechanical and corrosion properties; also, other types of coating techniques should be mentioned for comparison. Suggested reference: 10.1016/j.apsusc.2015.05.111.
2. In "2.1. Coating deposition," authors presented the coating process, but it should have included some images during the process.
3. SEM parameters are poorly written; please add more information.
4. In the XRD analysis, please add the ICDD codes for the compounds.
5. Please state which are the main applications for these coatings.
The rest is fine.
